# Effects of Maternal High-Fructose Diet on Long Non-Coding RNAs and Anxiety-like Behaviors in Offspring

**DOI:** 10.3390/ijms24054460

**Published:** 2023-02-24

**Authors:** Yuchen Zou, Qing Guo, Yidan Chang, Yongyong Zhong, Lin Cheng, Wei Wei

**Affiliations:** Child and Adolescent Health, School of Public Health, China Medical University, No. 77 Puhe Road, Shenyang North New Area, Shenyang 110122, China

**Keywords:** gestation, lactation, brain development, full-length RNA sequencing, Oxford Nanopore Technologies

## Abstract

Increased fructose intake is an international issue. A maternal high-fructose diet during gestation and lactation could affect nervous system development in offspring. Long non-coding RNA (lncRNA) plays an important role in brain biology. However, the mechanism whereby maternal high-fructose diets influence offspring brain development by affecting lncRNAs is still unclear. Here, we administered 13% and 40% fructose water to establish a maternal high-fructose diet model during gestation and lactation. To determine lncRNAs and their target genes, full-length RNA sequencing was performed using the Oxford Nanopore Technologies platform, and 882 lncRNAs were identified. Moreover, the 13% fructose group and the 40% fructose group had differentially expressed lncRNA genes compared with the control group. Enrichment analyses and co-expression analyses were performed to investigate the changes in biological function. Furthermore, enrichment analyses, behavioral science experiments, and molecular biology experiments all indicated that the fructose group offspring showed anxiety-like behaviors. In summary, this study provides insight into the molecular mechanisms underlying maternal high-fructose diet-induced lncRNA expression and co-expression of lncRNA and mRNA.

## 1. Introduction

In recent years, people have shown increasing preference for sweet foods in their diet, and sugar-sweetened beverage (SSB) demand is increasing [1]. A sweet monosaccharide, fructose, is the major form of high-fructose corn syrup (HFCS) [2], which is widely added to foods and SSBs [3,4]. A common phenomenon suggests that mothers exposed to such a diet environment are also at risk of excessive fructose intake. In America, pregnant women consume more than the recommended intake of SSBs, and fructose intake is out of balance [5,6]. A high-fructose diet can affect many biological processes and functions, including the nervous system [7,8]. Studies have shown that maternal high-fructose diet during gestation or lactation can cause insulin resistance and inflammation in the child’s brain [9,10], although the mechanisms have not been explored in depth. In our previous studies, maternal high-fructose diet could influence many biological processes related to brain development by changing transcription expression [11]. Anxiety-like behaviors have become a research hotspot in recent years. A strong link was identified between metabolic syndrome and mood disorders caused by high-fructose diet [12]. The repeated intake of high levels of fructose during adolescence has been proven to lead to stress response disorders and increase anxiety-like behaviors [13,14]. However, the effect of maternal high-fructose diet during gestation and lactation on offspring anxiety-like behaviors is still unclear.

Long non-coding RNA (lncRNA) has a length of 200 nt to 100,000 nt and does not encode proteins [15]. lncRNA can regulate gene expression by recruiting regulatory factors, which essentially change the spatial structure of adjacent mRNA genes [16]. lncRNAs can also affect the expression of distal genes by affecting trans-acting factors [17,18]. Consequently, lncRNA expression is correlated with the expression of their potential target genes [19,20], as shown by the lncRNA–target gene co-expression network. lncRNA expression is tissue specific and is especially high in brain tissues [21]. Some studies have shown that lncRNAs participate in nervous system development and function [22,23]. By affecting neurotransmitter transmission efficiency, brain derived neurotrophic factor (BDNF) content, and synaptic conduction [24,25], lncRNAs may regulate emotions [26]. However, lncRNAs are more difficult to detect and annotate, because the sequence conservation of lncRNA is lower than that of mRNA [27]. In nanopore sequencing, single-molecule electrical signals are sequenced in real time, making the accurate detection of lncRNA structural changes and expression changes possible [28,29].

In summary, we hypothesized that maternal high-fructose diet during gestation and lactation may change the expression of lncRNAs and lncRNA target gene co-expression and that this different expression may affect anxiety-like behaviors in offspring. To explore this, dams were exposed to 13% and 40% fructose water during gestation and lactation, and the hippocampus of their offspring was analyzed using Oxford Nanopore Technologies (ONT) full-length RNA sequencing. The co-expression networks of lncRNAs and mRNAs with large gene numbers were enriched to explore the relationship between lncRNAs and anxiety-like behaviors. Our findings reveal the key roles of lncRNAs and the interactions between protein-coding genes and lncRNAs in the effect of maternal high-fructose diet during gestation and lactation on offspring.

## 2. Results

### 2.1. Effect of Maternal High-Fructose Diet on Offspring Fasting Blood Glucose (FBG) and Fasting Insulin (FinS) during Gestation and Lactation

The gestation weight of dams in the fructose groups was significantly greater than that of dams in the control (Con) group (*p* < 0.01, *p* < 0.05) [11]. The weight of offspring in the 13% fructose (F13%) group was significantly greater than that of offspring in the Con group on postnatal day (PND) 30 and PND40 (*p* < 0.05); the weight of animals in the F13% and 40% fructose (F40%) groups were significantly greater than that of animals in the Con group on PND50 and PND60 (*p* < 0.01, *p* < 0.05) [11]. The 12 h FBG, 12 h FinS, and homeostasis model assessment results of PND21 and PND60 offspring in the fructose groups were all higher than those of age-matched offspring in the Con group (*p* < 0.01) [11]. 

### 2.2. Effects of Maternal High-Fructose Diet on Anxiety-Like Behaviors of Offspring during Gestion and Lactation

Frequency in central areas, duration of anxious state, duration of active state, number of standing, and trajectory in the open-field test were used to evaluate offspring anxiety-like behaviors. The frequency in central areas of animals in the fructose groups was significantly lower than that of offspring in the Con group (*p* < 0.05) (Figure 1A). The active state duration and anxious state duration of animals in the fructose groups were significantly longer than those of offspring in the F13% group (*p* < 0.01) (Figure 1B,C). Regarding the number of standing instances, the fructose groups displayed significantly higher values than the Con group (*p* < 0.01), and the F40% group also displayed significantly higher values than the F13% group (*p* < 0.01) (Figure 1D). Rats were more likely to move along the walls the higher the fructose concentration was (Figure 1E).

### 2.3. General Observation of Predicted LncRNAs

Using the inverse method, the coding potential calculator (CPC), the coding–non-coding index (CNCI), the coding potential assessment tool (CPAT), and Pfam were employed to predict the following lncRNAs: 1010 lncRNAs, 1232 lncRNAs, 882 lncRNAs, and 1100 lncRNAs, respectively. A total of 882 lncRNAs overlapped and were used for further analysis (Figure 2A). Of these lncRNAs, 57.9% were lincRNA; a total of 21.1% were sense lncRNAs; a total of 16.3% were antisense lncRNAs; and a total of 4.6% were intronic lncRNAs (Figure 2B). Regarding gene differential expression analysis, there were 181 (up-regulated, 105; down-regulated, 76) differently expressed mRNA genes in the control group versus the 13% fructose group (Con/F13%) group, 297 (up-regulated, 215; down-regulated, 64) in the control group versus the 40% fructose group (Con/F40%) group, and 374 (up-regulated, 204; down-regulated, 170) in the 13% fructose group versus the 40% fructose group (F13%/F40%) group. A total of 11 (up-regulated, 3; down-regulated, 8) differentially expressed lncRNA (DElncRNA) genes in the Con/F13% group, 19 (up-regulated, 9; down-regulated, 10) in the Con/F40% group, and 9 (up-regulated, 7; down-regulated, 2) in the F13/F40% group were found. These expression levels are shown below in a volcano map (Figure 2C).

We also compared the genomic features of lncRNAs and mRNAs in the three groups. There were 3460.5 bp of lncRNAs and 12,599 bp of mRNA in the Con group; a total of 3467 bp of lncRNAs and 12,599 bp of mRNAs in the F13% group; and a total of 3475.5 bp of lncRNAs and 14,362 bp of mRNAs in the F40% group. Further, lncRNAs were shorter than mRNAs in every group (Figure 3A). It was also apparent that lncRNAs had smaller open reading frames (ORFs) than mRNAs, with an ORF length of 201 bp versus 981 bp for mRNAs in the Con group, 981 bp for F13%, and 1053 bp for F40% (Figure 3C). The exons of lncRNAs were shorter than those of mRNAs. Further, the average exon length of lncRNAs was 639.5 bp, and that of mRNAs was 1633 bp in the Con group; the average exon lengths were 639 bp for lncRNAs and 1633 bp for mRNAs in the F13% group; and they were 639 bp for lncRNAs and 1741 bp for mRNAs in the F40% group (Figure 3B). The introns of lncRNAs were shorter than those of mRNAs. Further, the average intron length of lncRNAs was 2690 bp, and it was 10,494 bp for mRNAs in the Con group; the average intron lengths were 2697 bp for lncRNAs and 10,494 bp for mRNAs in the F13% group; and they were 2698.5 bp for lncRNAs and 12,224 bp for mRNAs in the F40% group (Figure 3D). In summary, lncRNAs were longer and possessed shorter exons and introns. Moreover, lncRNAs were also longer in the fructose groups.

### 2.4. Target Gene Identification, LncRNA–Target Gene Co-Expression Pattern Analysis, and Enrichment Analysis

The target genes were identified using cis and trans methods and are provided in Appendix A. These results were used to perform lncRNA–target gene co-expression analysis, and the complete networks are shown in Appendix A. We also combined the cis and trans results of DElncRNAs in the Con/F13% group and the Con/F40% group (Appendix A).

#### 2.4.1. Gene Ontology (GO) Enrichment

Clusters with more than ten genes were labeled in different colors in GO enrichment analysis. The cis lncRNA–target gene co-expression cluster with the highest number of genes in the F13% group was labeled in red, and the top terms were “positive regulation of transcription from RNA polymerase II promoter” in Biological Process (BP), “nucleus” in Cellular Component (CC), and “structural constituent of ribosome” in Molecular Function (MF). The next cluster was marked in blue, and the top terms were “positive regulation of transcription from RNA polymerase II promoter” (BP), “nucleus” (CC), and “ATP binding” (MF). The last one was marked in yellow, and the top terms were the same as the blue cluster (Figure 4A). The clusters obtained with the trans method were labeled in red and blue, and the top terms of the two clusters were the same as those obtained with the cis method (Figure 4C). The representative clusters of the F40% group contained fewer genes than the F13% group and were identified as red, blue, and yellow. To our surprise, the top terms of red and yellow clusters were the same as the blue cis cluster of the F13% group as follows: “response to drug” (BP), “nucleus” (CC), and “ATP binding” (MF). The results of the three clusters are provided in Figure 4E. Unfortunately, there was only one representative cluster in the trans results, and it was not sufficient for enrichment analysis. We also performed enrichment analysis on DElncRNA target genes, and the results are provided in Figure 5A. In the Con/F13% group, the top5 CC terms were “nucleus”, “nucleoplasm”, “neuronal cell body”, “extracellular vesicular exosome”, and “axon”. Among these CC results, some terms were related to brain development, such as “postsynaptic density”, “dendrite”, and “postsynaptic membrane”. The top terms for BP and MF were “positive regulation of transcription from RNA polymerase II promoter” and “ATP binding”, respectively. In the Con/F40% group, we paid greater attention to terms of BP, and the top five were “response to drug”, “intracellular signal transduction”, “neuron migration”, “positive regulation of GTPase activity”, and “protein ubiquitination”. “Neuron migration”, “brain development”, “hippocampus development”, and “memory” were the BP results related to the nervous system. The top terms of CC and MF were “nucleus” and “ATP binding”, respectively.

#### 2.4.2. Kyoto Encyclopedia of Genes and Genomes (KEGG) Pathway Enrichment

The KEGG pathway enrichment analysis also highlighted some pathways. Regarding the cis results of the F13% group, the top three terms of the red cluster were “Parkinson disease”, “Oxidative phosphorylation”, and “Amyotrophic lateral sclerosis”; the terms of the blue cluster were “Leukocyte transendothelial migration”, “Tight junction”, and “Renin-angiotensin system”; and the terms of the yellow cluster were “Ribosome”, “Wnt signaling pathway”, and “Ubiquitin mediated proteolysis” (Figure 4B, Appendix A). In the F40% group, there were also three colored clusters. The top three terms in the red cluster were “Melanoma”, “Breast cancer”, and “Gastric cancer”; the terms in the blue cluster were “Ribosome”, “Oxidative phosphorylation”, and “Parkinson disease”; and the terms in the yellow cluster were “Tight junction”, “Leukocyte transendothelial migration”, and “Cell adhesion molecules” (Figure 4D, Appendix A). Regarding the trans results, two clusters in the F13% group had enough genes for KEGG enrichment analysis, and the top three terms for the red cluster were “Thermogenesis”, “Huntington disease”, and “prion disease”, while the top three for the blue cluster were “Ribosome”, “Wnt signaling pathway”, and “Ubinquitin mediated proteolysis” (Figure 4F, Appendix A). For the DElncRNA target gene enrichment results, the top five pathways were “Oxytocin signaling pathway”, “MAPK signaling pathway”, “Hypertrophic cardiomyopathy”, “Dilated cardiomyopathy”, and “Cardiac muscle contraction” in the Con/F13% group; and “MAPK signaling pathway”, “Adrenergic signaling in cardiomyocytes”, “prion disease”, and “Osteoclast differentiation” in the Con/F40% group (Figure 5B, Appendix A).

In these KEGG enrichment results, we noted a term referring to “Dopaminergic synapses”. It was found in the cis blue and yellow clusters of the F13% group (*p*-values and enrichment factors: 0.1550, 1.9229; 0.5555, 1.2463, respectively), the trans blue clusters of the F13% group (0.5134, 1.4021), the cis red and yellow clusters of the F40% group (0.0771, 4.3419; 0.0545, 2.8042, respectively), and the DElncRNA target genes of F40% (0.0417, 6.1182). We performed a gene set enrichment analysis (GSEA) on this term to reveal the expression of all genes annotated to “Dopaminergic synapse”. The enrichment score (ES) peak values were all greater than zero in the Con/F13% and Con/F40% groups, which indicates that in the fructose groups, the up-regulated genes were dominant in “Dopaminergic Synapse” (Figure 6A).

### 2.5. Effect of Maternal High-Fructose Diet on Dopamine (DA) Changes of Offspring during Gestion and Lactation

According to the enrichment results of “Dopaminergic synapses”, we further tested the DA level in the serum of PND21 and PND60 offspring. On PND60, fructose group offspring had a significantly higher DA level than the Con group (*p* < 0.05) (Figure 6B). Moreover, we explored the expression level of dopamine receptor D1 (DRD1) and dopamine receptor D2 (DRD2), and these receptor proteins offructose groups were significantly up-regulated (*p* < 0.05) (Figure 6C). These results, along with results from the open-field test, confirm that maternal high-fructose diet may influence anxiety-like behavior in offspring.

### 2.6. Verification Tests of Sequencing and Enrichment Analysis Results

To ensure the validity of sequencing and enrichment analysis, some validation tests were carried out. We validated two enriched pathways, the “PI3K/Akt pathway” and the “AMPK pathway”, by testing the changes in core protein expression. The Western blot (WB) results showed no significant differences in PI3K, Akt, nor AMPK, while p-PI3K, p-Akt, and p-AMPK in the fructose groups were all significantly up-regulated compared with the Con group (*p* < 0.05) (Figure 7A). Regarding the validation of gene expression, we selected eight DElncRNA genes: ONT.13539 (*p*-value = 0.0003; log_2_ (fold change) = 1.1685), ONT.119 (0.0057; −0.9770), ONT.13715 (0.0067; −0.6060), and ONT.11765 (0.0077; −0.9760) in the Con/F13% group; ONT.11295 (0.0002; −0.6188), ONT.1222 (0.0014; 1.0059), ONT.5939 (0.0030; −1.3201), and ONT.252 (0.0045; 0.84278) in the Con/F40% group. The reverse transcription quantitative polymerase chain reaction (RT-qPCR) results are provided in Figure 7B. The expression of all genes was significantly different (*p* < 0.05) according to the comparison, and this expression difference was consistent with the sequencing results. We also predicted the Transcription factor binding site (TFBS) of these DElncRNA genes, as shown in Figure 7C. In addition, immunofluorescence (IF) was used to test the expression of BDNF, which is related to brain development. Clear distinctions were evident, and BDNF was reduced in the fructose groups (Figure 7D).

## 3. Discussion

Processing technology has made it easier for consumers to consume convenient and delicious processed foods [1], and these foods are enriched with HFCS [2]. Several studies have shown that eating high-fructose foods for a long time may affect stress response and anxiety-like behaviors [12,13,14]; however, the mechanism is still unclear. lncRNA plays a certain role in nervous system development and is also closely related to emotion regulation [24]. This study examined the effect of maternal high-fructose diets during gestation and lactation on offspring lncRNAs and anxiety-like behavior.

lncRNAs were predicted using ONT full-length RNA sequencing. Research has shown that the functional annotation of lncRNAs and the co-expression analysis of lncRNA target genes are powerful methods for the functional analysis of lncRNAs [19]. Our previous study suggested that maternal high-fructose diet during gestation and lactation can affect offspring neurodevelopment by affecting transcript and gene expression [11]. In this study, we also noted that “in utero embryonic development” was enriched in the blue cluster of the F40% group as found using the cis method. Embryo development is a complicated process. It involves the differentiation and growth of a variety of cells, including neuronal cells. Special lncRNAs are able to regulate genes that play key roles in embryonic development [30,31] and in the growth of neuronal cells via both the cis and trans regulation of coding genes [32,33], which is very similar to the role of transcription factors. lncRNAs are specifically expressed in neural stem cell differentiation and regulate the genes of transcription factors that are critical to neural stem cell self-renewal [34]. They can also influence neural stem cell differentiation through chromatin remodeling [35]. In our enrichment analysis of several co-expression clusters and DElncRNA target genes, we observed overlapping results regarding “neuronal cell body” and “negative regulation of neuron differentiation”. Further, “neuron migration” was enriched, and this term was also highlighted in our previous enrichment analysis results of differentially expressed transcripts (DETs) [11]. Neuron migration coordinates the formation of various brain structures during brain development [36]. Most neurons migrate from their birthplace to specific brain locations and then extend out of axons; neurodevelopmental dysregulation may occur when neuronal or axonal migration is disrupted [37]. This indicates that maternal high-fructose diet during gestation may seriously affect the development of the fetal brain.

Moreover, lncRNA might influence cognitive function in the central nervous system [38]. Our study found that “hippocampus development” and “memory” of DElncRNA target genes were enhanced. Memory and learning occur in the hippocampal region, and synapses are integral to these processes. Consistently, there were two terms that were enriched in the top ten list: “postsynaptic density” and “postsynaptic membrane”. Studies have shown that synaptic ncRNA-mRNA clusters were more abundant than those in total tissue homogenates [39,40]. Specific long non-coding RNAs can influence synaptic gene expression in cultured hippocampal neurons by interacting with splicing proteins [41,42]. Furthermore, antisense lncRNAs have been demonstrated to regulate neurite formation by regulating several key proteins involved in the process [43,44]. We also noted that “dendrite” was highlighted in DElncRNA target gene results. By influencing the translation and expression of some dendrite-related mRNA, ncRNAs control dendrite cell bodies [45,46]. In Smalheiser [40], the relationship between ncRNAs and learning and memory is further elaborated, in addition to these significantly enriched neuronal structures. In summary, maternal high-fructose diet during gestation and lactation can affect synapse function by changing lncRNA–target gene co-expression; as a result, the cognition of offspring is affected.

lncRNAs can also affect brain development. Further, “brain development” was found not only in lncRNA–target gene co-expression but also in DET enrichment results [11]. Allen Brain Atlas (ABA) [47] is a large-scale investigation based on gene expression in adult mice. A large number of non-coding RNAs detected in ABA were linked to specific neuroanatomical areas and neurons [48]. It provided convincing evidence that lncRNAs are strongly associated with brain development. Furthermore, lncRNAs have also been linked to neurodegenerative diseases [49,50], such as Alzheimer’s disease (AD), Parkinson’s disease (PD), and Huntington’s disease (HD), and these were also highlighted in our enrichment analysis. AD is the most common neurodegenerative disorder, and amyloid β (Aβ) is the key part that causes AD [51]. Some lncRNAs and their target genes were specifically expressed in the brain tissues, cerebrospinal fluid, and even blood of AD patients [52,53]. Moreover, some lncRNAs, such as Neuroblastoma differentiation marker 29, could influence the Aβ42: Aβ40 peptide ratio [54,55]. PTEN-induced kinase 1 (PINK1) is a protein that plays a crucial role in PD, which is a chronic neurological disease. It has been shown that silencing lncRNAs may affect PINK1 expression by regulating PINK1 encoding genes [56]. Other scholars have found that lncRNA expression was changed in the brain tissues of HD patients and HD mouse models [57].

To our surprise, “Cardiac muscle contraction”, “Arrhythmogenic right ventricular cardiomyopathy”, and “VEGF signaling pathway” were enriched. lncRNA can regulate chromatin and affect epigenetic inheritance by forming RNA-RNA, RNA-DNA, or RNA-protein complexes [58,59,60]. Heart disease has been linked to cognitive impairment, and the effects of the hippocampus on cognitive function extend to the circulatory system [61]. The hippocampus is also a potential sentinel site for ischemic lesions after resuscitation cardiac arrest [62,63]. Our research revealed that hippocampal function is affected in terms of signaling pathways, which can have cardiovascular effects. Simultaneously, research has pointed out that cardiac diseases were closely related to chromatin regulation, as chromatin-regulating factors could program cardiac gene expression to regulate cardiomyocyte development in embryos; control cell proliferation and differentiation in the neonatal period; and trigger gene recoding when the mature heart was stimulated [64]. lncRNAs have also became a key target in the treatment of cardiovascular diseases [60]. We also noted that the “AMPK signaling pathway” was highlighted in many of the top enrichment pathway results. AMPK used to be considered as an important regulator of heart energy metabolism; however, some scholars have pointed out that whether AMPK is a friend or foe is still unclear [65]. Many studies have shown that either abnormal activation or deactivation could cause cardiomegaly [66,67,68], and in our verification results, p-AMPK was significantly up-regulated. Both high and low expression of AMPK may affect neuroplasticity [69]; however, the mechanism is not clear at present. We also performed some verification experiments to ensure the accuracy of ONT full-length RNA sequencing. The “PI3K/Akt signaling pathway” was verified, and studies have shown that this pathway is up-regulated in anxiety-like behaviors [70,71,72]. A number of studies have shown that the PI3K/Akt signaling pathway is closely related to the learning and memory functions of the hippocampus and that it is a key pathway for the hippocampus to affect cognitive function [69,73]. Additionally, lncRNAs could regulate target genes to affect brain development, and BDNF has been widely recognized as a key neurotrophic factor involved in brain development [74,75,76]. Consistently, the expression of BDNF was decreased in fructose groups. 

Anxiety-like behaviors and their molecular mechanisms were investigated in experiments. On PND60, the open-field test was performed on offspring. Compared with the Con group, fructose group offspring were more active and restless in unfamiliar surroundings. The DA level in PND21 and PND60 offspring serum was also tested, and the results confirmed the open-field test results, i.e., the fructose groups had significantly higher DA levels than the Con group. The relationship between emotion and DA remains a hot topic, and one recognized view is that too much or too little of either could have adverse effects [77,78]. Studies have revealed that the decrease in dopamine transporter (DAT) may improve anxiety-related behaviors [79]; the inhibition of DA neurons is also necessary for antianxiety effects [80]; in the social isolation animal model, DA release and DA transporter activity were sustained and increased, and anxiety-like behaviors appeared [81,82]. In many of these enrichment analysis results, “Dopaminergic synapse” is stressed. The dopaminergic synapses regulate the release, diffusion, and uptake of DA [83,84]. The expression of all genes annotated to “Dopaminergic synapse” showed that the up-regulated genes contributed more. The fructose groups showed high levels of expression of DRD1 and DRD2 receptors, which play a key role in anxiety regulation [78]. These results further confirm that maternal high-fructose diet could influence offspring anxiety-like behaviors by influencing offspring dopaminergic synapses and DA content, and this effect may be caused by transcription differential expression, as well as lncRNAs that regulate gene encoding and expression.

## 4. Materials and Methods

### 4.1. Animals and High-Fructose Diet

Our study included 18 healthy two-month-old female and male SD rats raised at Centre for Experimental Animals at China Medical University (Shenyang, China). The rats were raised in a constant environment (20–25 °C, 50–65%), with 12 h light and 12 h dark cycles. After one week of adaptive feeding, the rats were mated (♀:♂ = 2:1). Gestation day (GD) 0 is the day when the vaginal plug was found. On GD0, the dams were randomly divided into Con (n = 6), F13% (n = 6) [85,86], and F40% (n = 6) [87,88]. PND0 is the day when a dam gave birth to offspring. From GD0 to PND21, the three groups drank different-concentration drinks with 0 g/mL, 13 g/mL, and 40 g/mL D-fructose (Solibao, Beijing, China) and were fed uniform fodder used in feeding centers. Offspring were separated and reared according to sex on PND21. Then, all the offspring started the same normal diet until they were PND60. We weighted dams and offspring, tested the offspring 12 h FBG and 12 h FinS on PND21 and PND60. Rats were sacrificed with diethyl on PND60, and materials were stored at −80 °C. Every effort was made to minimize the suffering of animals. Figure 8 illustrates an intuitive experimental route. 

### 4.2. Open-Field Test

In a novel environment, an open-field test can evaluate the autonomous behavior, inquiry behavior, and tension of animals. The experimental device consisted of an open-field chamber and a data recorder (Noldus EthoVison XT; Wageningen, The Netherlands). The dark chamber, of 100 cm in length × 100 cm in width × 60 cm in height, was divided into 16 virtual squares of 25 cm × 25 cm, and the 4 squares surrounding the center were the central regions. The test was performed using six rats (♀:♂ = 1:1) from each group at 20:00–22:00 on PND60. We placed each rat in the same corner, and it could freely explore the strange surroundings for 5 min. Simultaneously, the system traced the movement of rats and calculated the speed and frequency of young rats; a rat with a speed of over 50 cm/s was considered anxious, and one with a speed between 30 and 50 cm/s was considered active. To avoid residual information, the chamber was cleaned with 95% alcohol between trials. The speed level, frequency in the central area, number of standing instances, and trajectory were used to measure the offspring anxiety-like behavior.

### 4.3. ELISA

From each group, six rats (♀:♂ = 1:1) were randomly selected, and their blood was kept at room temperature for 2 h and centrifuged at 12,000 rpm for 20 min at 4 °C. The upper serum was used for ELISA. We assayed the DA level using Rat DA ELISA Kit (Enzyme-linked Biotechnology, Shanghai, China). Prepared as above, aliquots of 50 μL of DA standards (0, 0.05, 0.1, 0.2, 0.4, 0.8, and 1.6 ng/mL) were added to standard wells, while 50 μL of the sample was added to test sample wells. Each well was then incubated for 30 min at 37 °C with 50 μL of HRP-conjugated reagent after gently mixing. Chromogen solution A and solution B were added to each well after incubating and washing, and the plate was kept at 37 °C for 15 min in the absence of light. In the last step, 50 μL of stop solution was added to each well. We read the absorbance at 450 nm using a microplate reader (H1MD; Cube Biotech, Monheim, Germany). A curve was generated, and the DA levels of the samples were calculated.

### 4.4. Preparation of Basic RNA Sequencing Data

Full-length nanopore RNA sequencing was conducted according to the protocol provided by Oxford Nanopore Technologies (Oxford, UK), including sample quality detection, cDNA library construction, library data polishing, and sequencing. Eight hippocampi from PND60 offspring were randomly selected in each group (♀:♂ = 1:1). Total RNA from the tissues was isolated, quality-tested, reverse-transcribed, and subjected to magnetic bead purification, and the final cDNA libraries were run on the PromethION platform (Oxford Nanopore Technologies plc, Oxford, UK) at Biomarker Technology Company (Beijing, China). Raw reads were filtered with a minimum quality score of 6 and a minimum length of 500 bp to avoid affecting the subsequent analysis. From the clean data, full-length non-chimeric (FLNC) transcripts were identified by detecting primers at both ends of the reads (Appendix A). The full-length sequence was compared with the reference genome using minimap2 software (version 2.16). The genome was ENSEMBLE (Rnor_6.0_release95). After clustering using the comparison information, the consistency sequence was obtained using pinfish software (version 0.1.0). Finally, consensus sequences were mapped to the reference genome using minimap2. Mapped reads were further collapsed using the cDNA Cupcake package with min-coverage = 85% and min-identity = 90%. The 5′ difference was not considered when collapsing redundant transcripts.

### 4.5. LncRNA Prediction and Differential Expression LncRNA Identification

As lncRNA does not encode proteins, it is necessary to determine whether lncRNA has coding potential by screening transcripts for coding potential and find transcripts that do not have coding ability, so that the transcripts can be identified as lncRNA. CPC [89], CNCI [90], CPAT [91], and Pfam [92] were used to predict the transcripts with coding potential; then, the transcripts without coding potential were obtained using the inverse extrapolation method. The indexes we used were as follows: for CPC and CNCI, non-protein-coding RNA with score < 0; for CPAT, non-protein-coding RNA with score > 0.38; for Pfam, domain screening condition e-value < 0.001. lncRNA analysis was conducted using the intersection of non-protein-coding transcripts identified using the above four methods. Moreover, the counts per million of all transcripts in each sample were calculated using the EdgeR package (version 3.8.6). lncRNAs with a *p* < 0.05 and fold change ≥ 1.5 were DElncRNAs. 

### 4.6. Target Gene Prediction and LncRNA–Target Gene Co-Expression Network Construction

We used the cis and trans prediction methods to predict lncRNA sequences. By using the cis method, lncRNAs control the expression of adjacent genes. The target genes were defined based on the location of lncRNAs and mRNAs on chromosomes. Genes within the 100 kb range of lncRNA were cis target genes. Trans regulation relies on base complementary pairing between lncRNA and mRNA, and the LncTar target gene prediction tool is exclusively used to predict trans target genes. Then, these correlations were shown as co-expression networks using Cytoscape software (version 3.9.1).

### 4.7. Functional Enrichment Analysis

We selected co-expression clusters with many genes from the network and performed GO enrichment analysis, KEGG pathways enrichment analysis, and GSEA on these clusters. The GO database was established by the Gene Ontology Consortium and is applied to all species. It also describes the functions of proteins and genes with a standard vocabulary system. This database consists of BP, CC, and MF main branches. The GO-seq R packages, based on Wallenius noncentral hypergeometric distribution, were used to perform GO enrichment analysis. KEGG includes the current molecular network interactions, such as icon channels and complexes. KEGG enrichment analysis was performed using KOBAS, and statistical indicators used the proportion of annotated transcripts and enrichment factors. GSEA can understand the expression trend of genes in specific functional gene sets and whether the expression trend has any statistical significance [93]. In the resulting graph, the green line represents the running gene ES, and the ES peak is used to reveal the core genes under this gene cluster; hits represent each gene under this gene set; rank distribution is shown at the bottom. This analysis was implemented in the R-GSEA program, and the gene sets with NES > 1 or < −1 (*p*-value < 0.05) were the leading-edge subsets.

### 4.8. IF

Brain tissues from six rats (♀:♂ = 1:1) were randomly selected in each group. After 36 h of fixation, the tissues were embedded in paraffin blocks and cut into 6 μm thick sections. Complete hippocampus sections were baked at 60 °C for 2 h. Antigen repair was carried out for 15 min, followed by 30 min of preincubation in 10% normal goat serum. Then, the sections were incubated with rabbit anti-BDNF (diluted to 1:100; ABclonal, Wuhan, China) at 4 °C overnight. The sections were then incubated with the second antibody (Cy3 goat anti-rabbit; diluted to 1:800; ABclonal, Wuhan, China) at room temperature for 2 h. The sections were sealed with an antifade mounting medium in a dark environment and observed at ×200 magnification under a forward fluorescence microscope (80i; Nikon Corporation, Tokyo, Japan).

### 4.9. WB

In each group, six protein homogenates (♀:♂ = 1:1) were randomly selected. After centrifuging at 12,000 rpm for 10 min at 4 °C, the supernatant was separated from the homogenate and assessed using BCA Protein Assay Kit (Beyotime Biotechnology, Shanghai, China). We adjusted the concentration of protein to 3 μg/μL with phosphate-buffered saline (PBS) and 5× SDS loading buffer (Beyotime Biotechnology, Shanghai, China), and sample degeneration was performed at 100 °C for 5 min. Vertical electrophoresis was performed under the constant voltage of 120 V for 75 min; then, the proteins were transferred onto polyvinylidene difluoride membranes (Thermo Fisher Scientific, Waltham, MA, USA) under the constant voltage of 100 V for 75 min. The membranes were cut at the location of the target proteins following blocking in 5% skimmed milk for 90 min. Then, these cuttings were incubated with the following primary antibodies: rabbit anti-DRD2, rabbit anti-PI3K catalytic subunit alpha, rabbit anti-Akt 1, and rabbit anti-AMPK (diluted to 1:1000; ABclonal, Wuhan, China); rabbit anti-*p*-Akt (Ser473), rabbit anti-*p*-AMPKα (Thr172), and rabbit anti-β-actin (diluted to 1:1000; Cell Signaling Technology, Boston, MA, USA); rabbit anti-*p*-PI3K (Tyr607) (diluted to 1:500; Affinity Biosciences, Changzhou, China); rabbit anti-DRD1 (diluted to 1:1000; Proteintech, Wuhan, China). After incubation with the primary antibody at 4 °C overnight, the second antibody was incubated with HRP secondary antibody (goat anti-rabbit; diluted to 1:5000; ABclonal, Wuhan, China) at room temperature for 2 h. Chemiluminescence Western Blot Kits (Beyotime Biotechnology, Shanghai, China) were used to reveal the proteins using Tanon-5200 (Tanon, Shanghai, China). For each protein, β-actin was analyzed as quality control.

### 4.10. RT-qPCR

Total RNA was extracted from six rats (♀:♂ = 1:1, randomly selected in each group) using TRIzol (Vazyme, Nanjing, China), and RNA purity and concentration were evaluated using Nanodrop ND-2000 Spectrophotometer (Thermo Fisher Scientific, Waltham, MA, USA). The purity of all samples ranged from 1.9 to 2.2. RNA was dissolved in 0.1% diethyl pyro-carbonate (Vazyme, Nanjing, China) and transcribed into cDNA with Reverse Transcription Kit (HiScript III RT SuperMix for qPCR with gDNA wiper; Vazyme, Nanjing, China). The sequence of lncRNAs was provided by Biomarker Technology Company, and the primers were synthesized and produced by Sangon Biotech (Shanghai, China) (Appendix A). lncRNA expression was quantified with PCR using qPCR Kit (ChamQ Universal SYBR qPCR Master Mix; Vazyme, Nanjing, China). PCR was performed with QuantStudio 6 Flex Real-time System (Thermo Fisher Scientific, Waltham, MA, USA) according to the following curve: denaturation at 95 °C for 30 s; extension at 95 °C for 10 s; annealing at 60 °C for 30 s; finally, dissociation at 95 °C for 15 s, at 60 °C for 1 min, and at 95 °C for 15 s. We used β-actin (Sangon Biotech, Shanghai, China) as the control and calculated gene relative expression with the 2^−ΔΔCt^ method.

### 4.11. TFBS Prediction

TFBStools was used to predict TFBS in the promoter region of specific target genes, and the potential promoter was defined as a region of about 1 kb upstream of the gene. The reference TF motif database was JASPAR.

### 4.12. Statistical Analysis

All statistical analyses were performed using SPSS 21.0 software (IBM SPSS, Inc., Chicago, IL, USA). Unpaired Student’s *t*-test or one-way analysis of variance (ANOVA) was performed between groups. After ANOVA analysis, equal variance with least significant difference (LSD) and Student–Newman–Keuls (S-N-K), and unequal variance with Dunnett’s T3 were performed. *p* < 0.05 indicated that the difference was statistically significant. Some data were expressed as means ± SEMs. All graphs were created using GraphPad Prism 8.0 software (GraphPad Software, San Diego, CA, USA).

## 5. Conclusions

Maternal high-fructose diet during gestation and lactation changed the expression of lncRNAs and their target genes in the hippocampus of offspring, and this differential expression further affected multiple physiological functions, especially items related to brain development. In the enrichment analysis results, “dopaminergic receptors” was enriched, and our animal behavior and molecular experiments also confirmed that offspring showed anxious-like behaviors. Our study also suggests a relationship between lncRNAs and emotion regulation.

## Figures and Tables

**Figure 1 ijms-24-04460-f001:**
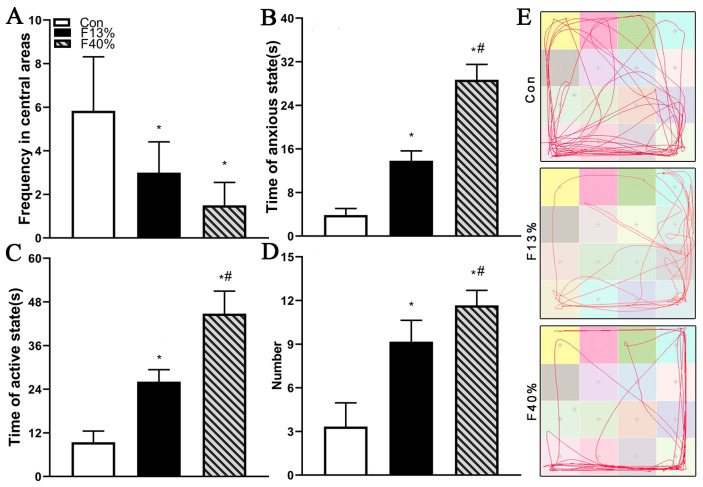
(**A**–**D**) Frequency in central areas (**A**), duration of anxious state (**B**), duration of active state (**C**), and number of standing (**D**) of animals in three groups. (**E**) The representative trajectories of animals in three groups. Note: Con means control group; F13% means 13% fructose group; F40% means 40% fructose; in every group, n = 6. Each bar represents mean ± SEM. * *p* < 0.05 (vs. control group); # *p* < 0.05 (vs. 13% fructose).

**Figure 2 ijms-24-04460-f002:**
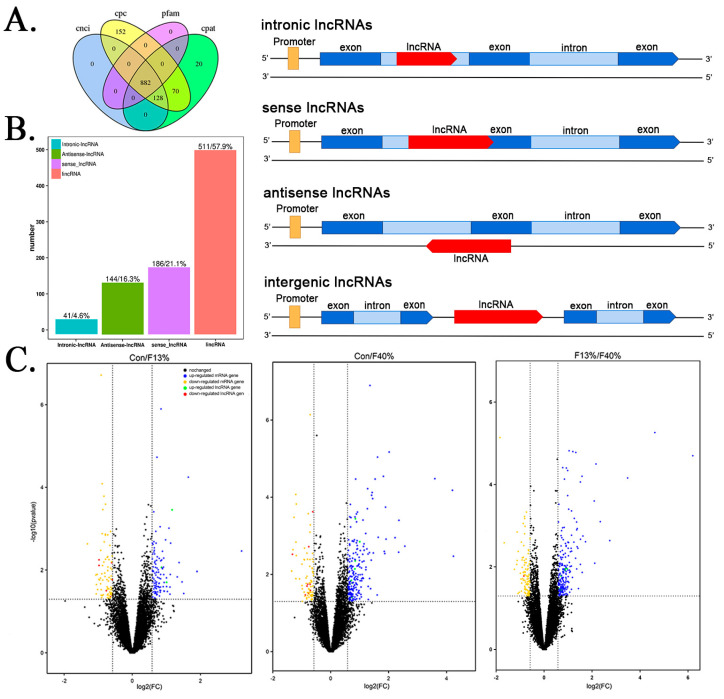
Overview of lncRNAs and different structures and expression between long non-coding RNAs (lncRNAs) and mRNAs. (**A**) Venn diagram of four lncRNA screening methods. (**B**) Classification of lncRNAs based on their genomic location. (**C**) Volcano map of the differentially expressed mRNAs and lncRNAs in the control group compared with 13% fructose group (Con/F13%), control group compared with 40% fructose group (Con/F40%), and 13% fructose group compared with 40% fructose group (F13%/F40%).

**Figure 3 ijms-24-04460-f003:**
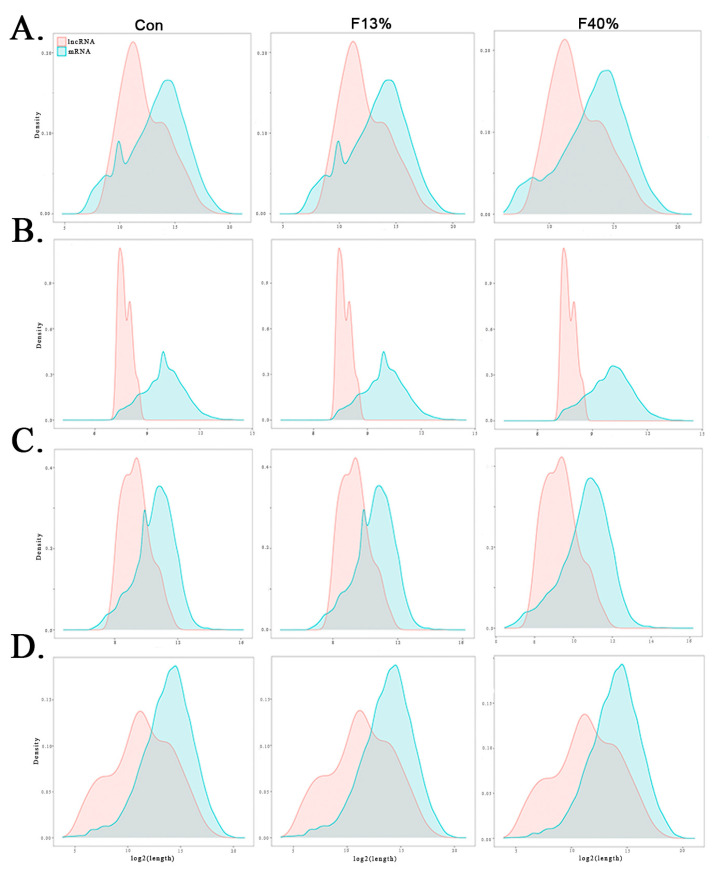
Structures and expression of long non-coding RNAs (lncRNAs) and mRNAs. (**A**) Full-length distribution of lncRNAs and mRNAs. (**B**) Number of exons per transcript. (**C**) Open reading frame size distribution of lncRNAs and mRNAs. (**D**) Number of introns per transcript.

**Figure 4 ijms-24-04460-f004:**
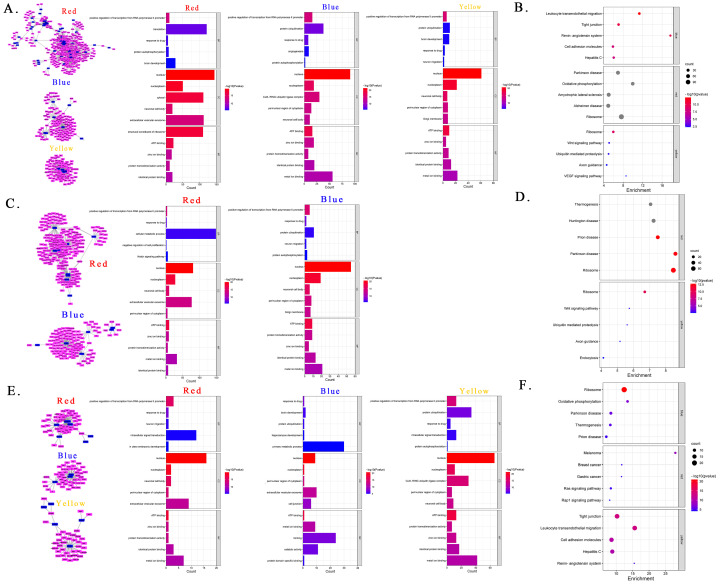
Gene Ontology (GO) and Kyoto Encyclopedia of Genes and Genomes (KEGG) enrichment analyses based on lncRNA–target gene co-expression clusters in fructose groups. (**A**,**B**) Enrichment analysis of three clusters in F13% using the cis method. (**C**,**D**) Enrichment analysis of two clusters in F13% using the trans method. (**E**,**F**) Enrichment analysis of three clusters in F40% using the cis method. Note: In the GO enrichment analysis results, the ordinate represents the name of GO terms; the abscissa represents the gene count; and the color represents the −log10 (*p*-value) of GO terms. In the KEGG pathway enrichment analysis, each circle represents a KEGG pathway; the ordinate represents the pathway name; the abscissa represents the enrichment factor; the circle color indicates the *p*-value; and the circle size indicates the number of enriched genes in the pathway.

**Figure 5 ijms-24-04460-f005:**
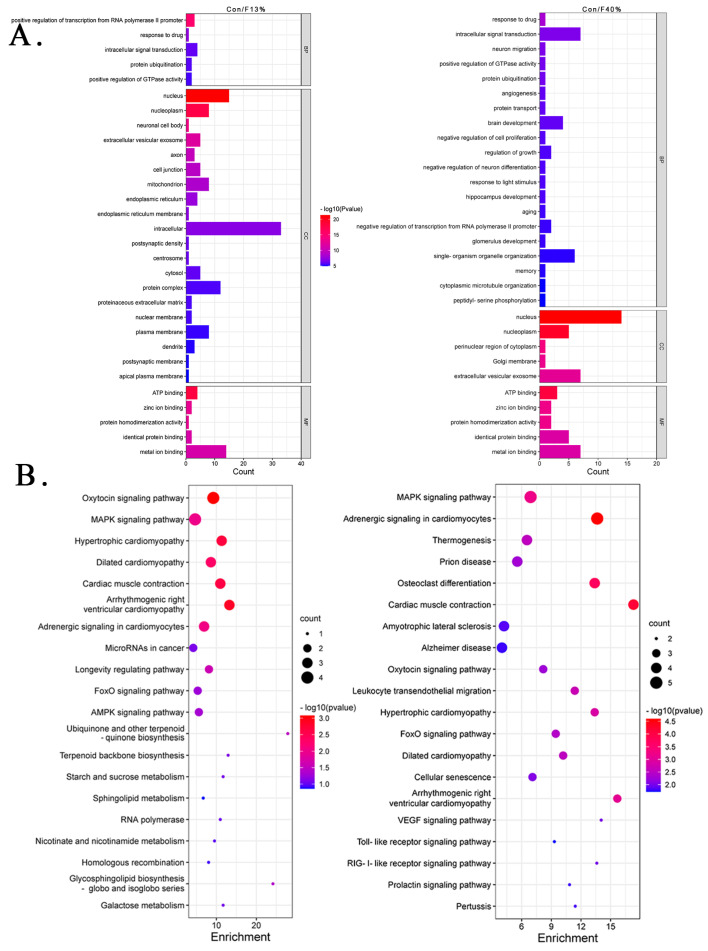
GO and KEGG enrichment analysis based on differentially expressed lncRNA (DElncRNA) target genes. (**A**) The top 20 GO enrichment analysis results of Con/F13% and Con/F40%. The ordinate represents the name of GO terms; the abscissa represents the gene count; and the color indicates the −log10 (*p*-value) of GO terms. (**B**) The top 20 KEGG pathway enrichment analysis results of Con/F13% and Con/F40%. Each circle represents a KEGG pathway; the ordinate represents the pathway name; the abscissa represents enrichment factor; the circle color indicates the *p*-value; and the circle size indicates the number of enriched genes in the pathway.

**Figure 6 ijms-24-04460-f006:**
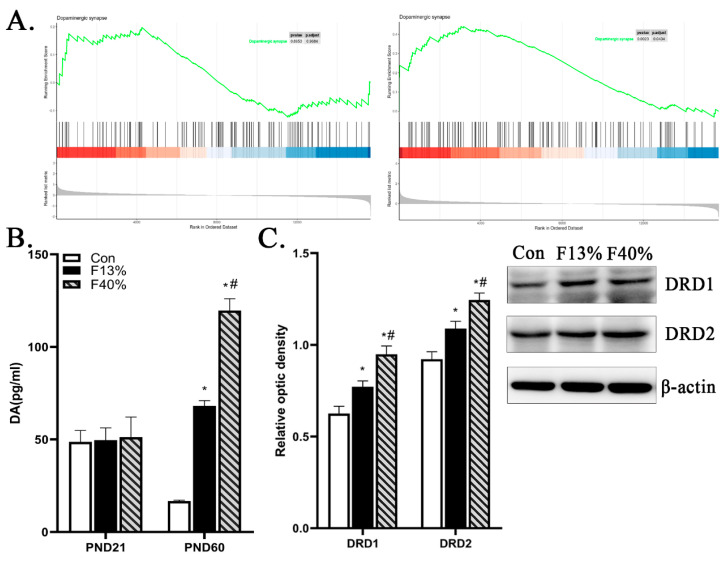
Changes in dopamine (DA) and dopaminergic synapse. (**A**) GSEA results of “dopaminergic synapse” of Con/F13% and Con/F40%. The abscissa represents the location of the ranked transcript set, and the ordinate indicates log_2_ (fold change). The bottom part of the graph is the sequence quantity distribution, and the green line in the top part is the transcript enrichment score. (**B**) The DA level of PND21 and PND60 offspring. (**C**) The representative immunoblots and densitometry analysis of DRD1 and DRD2. Note: Every group n = 6; each bar represents mean ± SEM; * *p* < 0.05 (vs. control group); # *p* < 0.05 (vs. 13% fructose). Con, control group; F13%, 13% fructose group; F40%, 40% fructose group.

**Figure 7 ijms-24-04460-f007:**
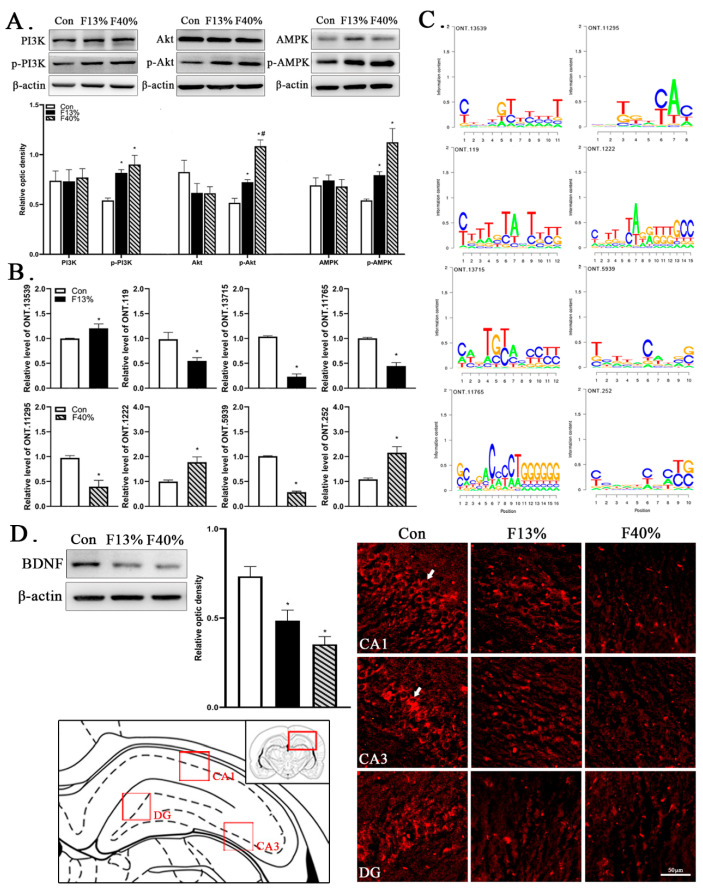
Transcriptome sequencing validation experiments and Transcription factor binding site (TFBS) prediction. (**A**) Representative immunoblots and densitometry analysis of PI3K, p-PI3K, Akt, p-Akt, AMPK, and p-AMPK expression levels in the total hippocampus. (**B**) Quantitative expressions of eight DElncRNAs. (**C**) TFBS of eight DElncRNAs. (**D**) Representative microscopic images of BDNF (red) IF staining in hippocampus CA1, CA3, and DG regions. White arrows indicate representative specific expression. Note: Every group n = 6; each bar represents mean ± SEM; * *p* < 0.05 (vs. control group); # *p* < 0.05 (vs. 13% fructose). In microscopic images, scale bar = 50 μm; magnification = ×200. Con, control group; F13%, 13% fructose group; F40%, 40% fructose group.

**Figure 8 ijms-24-04460-f008:**
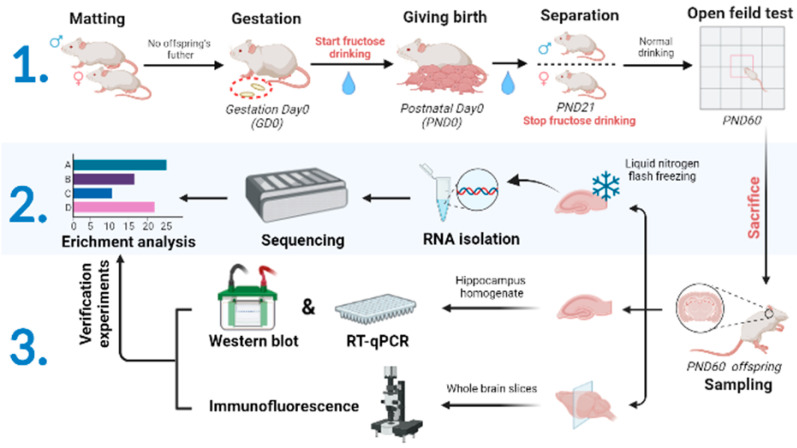
Animal model construction and experiment pipeline. The first step was the animal modeling process, the second was the sequencing analysis process, and the third was the experimental verification process.

## Data Availability

Data can be made available upon request.

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
