# Peer review of "Effects of Maternal High-Fructose Diet on Long Non-Coding RNAs and Anxiety-like Behaviors in Offspring"

_ijms, 2023, doi:10.3390/ijms24054460_

Round 1

Reviewer 1 Report

High fructose corn syrup is commonly used in the food and beverages world-wide. The maternal high-fructose diet during gestation and lactation has been demonstrated to affect many biological processes and functions in offspring, including nervous system development.  However, whether and how maternal high-fructose diet led to anxiety-like behaviors in offspring is still unclear. Zou et al. used a maternal high-fructose diet rat model to demonstrate the expression changes of lncRNAs and their potential target genes, which may underlie the mechanisms of high-fructose diet-induced anxiety-like behaviors. This is an interesting study; however, I have some suggestions and points of clarification below.

1.     How were the doses of 13% and 40% fructose water chose? Is it relevant to human data?

2.     In Result Section 2.1, FBG, FinS, and HOMA data have been published in Ref 11. The findings should only be described briefly in this manuscript, not in details with its own section. In parallel with this, methods for FBG, FinS, and HOMA should be removed from Method Section 5.1 and 5.3.

3.     Figure 1 is very confusing. The panels are not matched with the text and legends.

4.     In line 141-142, the summary statement is not consistent with the results described. Please check.

5.     The resolution for Figure 3 is too low and it is impossible to view the details.

6.     Please add tables and list all lncRNAs and their target genes used for GO and KEGG analysis in the manuscript. The expression changes of each lncRNA and target gene as well as their False discovery rate (FDR) should be included.

7.     Why did authors use P < 0.05, not FDR < 0.05, to identify DElncRNAs and DEGs?? For RNA-seq, FDR should be a more accurate method to identify differentially expressed genes.

8.     In Figure 4, please use high resolution pictures for panels A, C, and E.

9.     Please add a table and list the genes in Dopaminergic synapses pathway, including gene expression changes and FDR.  

10.  In Figure 7C, it is not clear whether these are TF binding sites on the promoters of DElncRNA genes? If so, what are these transcription factors that regulate DElncRNA expression?

11.  The authors demonstrated changes of BDNF expression in Figure 7D. Does BDNF a target gene of DElncRNA identified in the study? If so, which DElncRNA?

12.  Are any DElncRNAs identified in the current study also conserved in human? If so, which ones?

13.  In discussion section, the authors used a paragraph to discuss cardiac-related pathways (lines 340-353). Please explain why these pathways are important in hippocampus. Additionally, please discuss briefly the importance of PI3K/Akt pathway and AMPK pathway in hippocampal function.

Reviewer 2 Report

The present work from Zou et. al. "Effect of maternal high-fructose diet on long non-coding RNAs 2 and anxiety-like behaviours in offspring", shows an interesting approach to correlate lncRNAs with brain function and anxiety. It is a very interesting topic and a well-established experiment. Although some considerations must be addressed:

The most significant problem remains in the material and methods, especially within sequencing and bioinformatics approaches. Nanopore technology is a relatively new methodology when compared to other sequencing technologies. 

the session "Preparation of basic RNA-sequencing data" must be deeply described, there is crucial information missing, and almost no reference to methods, for example, lncRNAS can have from 200nt, and the threshold cut filtered was 500nt, why? A lncRNA with 400nt is biologically inert? Which version of the genome was used? How were new lncRNAs predicted? Why none of the most known lncRNAs were not mentioned?  Furthermore, on the session "LncRNA prediction and differential expression lncRNA identification" was mentioned PFAM, CPC, CNCI, and CPAT, those are tools for the calculation of coding potential, how the new lncRNAs were predicted?  The noncode platform was consulted? 

In the session "Target genes prediction and lncRNA-mRNA co-expression network construction", what does " R > 0.9 (positive) or R < 0.9 (negative), actually means? the p-value was considered? Also, why Pearson correlation method was used? In what type of data was used, and which package was used to perform? There are other methods to perform co-expression analysis why do you choose to use just Pearson correlation, pearson correlation test is usually used for normally distributed data, how was the data normalized?

The "Functional enrichment analysis" and "GSEA", why it was separated if GESEA stands for Gene Set Enrichment Analysis?

The contrast between 13% and 40% should be mentioned or shown, it was tested, if not why?

The results are well presented, but some images are only possible to see in the supplementary files with duplicated files.

The discussion session must be improved is too short for the size of the results. 

Data Availability Statement says: "Data are contained within the article"

Where? There are primer sequences, but there is no sequence of the lncRNAs, is the RNA-seq raw data going to be deposited somewhere?

Where is the count table?  

The work has great potential, but unfortunately in the present format I can not recommend it for publication, once those problems are addressed I would be happy to re-evaluate. 

Round 2

Reviewer 2 Report

Zhou et. al. somewhat answered most of the questions, although issues must be addressed. 

Results:

"2.3 General observation of predicted lncRNA" lines 96-109, must be reviewed, CPC, CPAT, and CNCI are mentioned as predictors of lncRNAs. 

Also, this is related to questions 1 and 2, The author's answer is not sufficient, the ENSEMBL and NCBI are genome banks, but which genome version has been used is not mentioned. Furthermore, the authors used  PHYZOME to check for the presence of lncRNA,  why? PHYZOME is a plant, primarily, a genome resource, why NONCODE wasn´t checked?  This part still needs improvement and results associated with this methodology also must be checked.   

Question 4, GSEA is a method for Functional enrichment analysis, also KEGG is encyclopedia, they must be part of the same result session. GSEA is  a method of Functional enrichment analysis published on https://doi.org/10.1073/pnas.0506580102, it is also the name of the software which performs the analysis and helps to produce graphical results. Although others software also can use GSEA in the background.  In summary, this part of the manuscript must be addressed and improved again.

The Point5  is :

"

Point 5: The contrast between 13% and 40% should be mentioned or shown, it was tested, if not why?

Response 5: Thank you for your valuable comments. We did not set 13%/40% group when designing the bioinformatics analysis, focused on comparing the difference between the high-fructose diet and the normal diet. However, in the verification of molecular experiments, some experimental results suggest that there is also some difference between 13% group and 40% group, which is beyond our expectation. We have realized that this is a meaningful finding that has been applied to follow-up studies.

"

It means that the authors have explored the results and will show then in another paper?

From my point of view, this result must be shown in this work, at least the results of differential gene expression analysis are part of this work, the present work is incomplete without that contrast. 

 Point 7

"

Point 7: The discussion session must be improved is too short for the size of the results.

Response 7: Thank you for your valuable comments, we have expanded the discussion session. Because there are too many modifications, we cannot list them all here. You can view the modifications in “Track Changes” function.

"

I had acces only to a pdf file and was not possible to see significant improvements on discussion. 

Once again, the work is meaningfull and interesting, but methods are not clear enough in a way that can make the result scientifically sound. The problems mentioned are fairly manageble. 
